# Living kidney donation in a developing country

**Chandni Dayal**[1]*, **Malcolm Davies**[1], **Nina Elisabeth Diana**[1], **Anthony Meyers**[1,2]

**1** Division of Nephrology, Department of Internal Medicine, University of the Witwatersrand, Johannesburg, South Africa, **2** National Kidney Foundation, Johannesburg, South Africa

* chandnidayal@gmail.com

## Abstract

### Background

Living kidney donation has been advocated as a means to ameliorate the chronic shortage of organs for transplantation. Significant rates of comorbidity and familial risk for kidney disease may limit this approach in the local context; there is currently limited data describing living donation in Africa.

### Methods

We assessed reasons for non-donation and outcomes following donation in a cohort of 1208 ethnically diverse potential living donors evaluated over a 32-year period at a single transplant centre in South Africa.

### Results

Medical contraindications were the commonest reason for donor exclusion. Black donors were more frequently excluded (52.1% vs. 39.3%; p<0.001), particularly for medical contra-indications (44% vs. 35%; p<0.001); 298 donors proceeded to donor nephrectomy (24.7%). Although no donor required kidney replacement therapy, an estimated glomerular filtration rate below 60 ml/min/1.73 m$^2$ was recorded in 27% of donors at a median follow-up of 3.7 years, new onset albuminuria >300 mg/day was observed in 4%, and 12.8% developed new-onset hypertension. Black ethnicity was not associated with an increased risk of adverse post-donation outcomes.

### Conclusion

This study highlights the difficulties of pursuing live donation in a population with significant medical comorbidity, but provides reassurance of the safety of the procedure in carefully selected donors in the developing world.

**Data Availability Statement:** All data underlying this study is available on the WIReDSpace repository (https://doi.org/10.54223/uniwitwatersrand-10539-32821).

**Funding:** The authors received no specific funding for this work.

**Competing interests:** The authors have declared that no competing interests exist.

## Introduction

The increasing global prevalence of kidney failure (KF) contributes significantly to the burden of chronic disease in developing countries [1, 2]. It is estimated that by 2030 more than 70% of patients with KF will reside in middle- and low-income countries where access to kidney replacement therapy (KRT) is limited; at present, less than one-fifth of the affected patient population in Africa has access to KRT [3, 4]. In underdeveloped parts of the continent, a lack of infrastructure and other socioeconomic limitations further deter access to all modalities of KRT [4].

South Africa faces a unique set of challenges in addressing this deficiency in KRT availability. Gross socioeconomic inequality, a legacy of the country's colonial and apartheid past, manifests itself in a discordant healthcare system comprised of a well-resourced private sector funded by medical insurers which caters to the advantaged, whilst the bulk of the population depends on a resource-limited state-funded public sector [5, 6]. The significant burden of communicable disease which afflicts the latter, notably the country's inter-related human immunodeficiency virus (HIV) and tuberculosis epidemics, demand a disproportionate quantum of the public sector healthcare resources, reducing funding for the widespread provision of KRT [6, 7]. As a result, dialysis access in the South African public health sector is rationed according to transplant eligibility [7–10]. The fortunate few receiving KRT in the public sector are fully funded by the public health authority, without incurring out-of-pocket expenses.

There are currently seven accredited centres each in the state and private sectors that offer kidney transplantation (KT) services in the country. These centres are distributed across four provinces and serve a total of 30 state-run dialysis units and a further 228 private sector dialysis facilities [5]. The present national KT rate of 6.4 per million population (pmp) in South Africa is on a downward trend, and is well below the KT rates of other upper-middle-income countries [2, 5, 8, 11]. According to South African Renal Registry data, the overall KT rate in 2019 was 4.8 pmp in the public sector and 15.2 pmp in the private sector, of which 58% were procured from deceased kidney donors [5]. Without emergent intervention to expand the availability of transplantation to the South African population, already strained state-funded KRT programmes are at risk of collapse [7]. Living kidney donation (LKD), which has been shown to be a cost-effective strategy to meet the growing demand for sustainable KT in the developing world, offers the potential to increase transplantation rates [1, 2, 9, 12].

LKD requires that healthy individuals endure a major surgical procedure devoid of any direct self-benefit [12]. The numerous advantages of living donor transplantation for the recipient must therefore be carefully balanced against immediate and long-term donor safety [13–16]. Data suggests that up to two-thirds of potential living donors fail to complete the donation process [17–21]. Understanding the reasons for non-donation is required to identify possible modifiable barriers for intervention in order to augment living donation rates [17, 20]. Furthermore, informed consent for donation requires communication of the potential risks to the donor which remain unclear, particularly for donors of Black ethnicity who seem to bear the greatest risk of adverse outcomes [22–26]. Whereas older reports of long-term donor follow-up suggested that the risks of donor nephrectomy were of limited clinical significance, emerging developed-world data comparing appropriately matched controls, suggests poorer long-term survival and an elevated risk of KF in living donors [27–33].

There is a paucity of data regarding the living donor selection process and outcomes of post-donation follow-up in demographically diverse populations in the developing world. The present study was undertaken to evaluate living donation in the South African context with the aims of characterising reasons for non-donation, examining morbidity and mortality

following donation, and identifying whether any differences exist in outcomes between demographic subgroups in this setting.

## Methods

Written ethics approval for this study was granted by the Human Research Ethics Committee (Medical) of the University of the Witwatersrand (clearance certificate number M150923). This approval permitted a folder review of all patients assessed within the defined study period. Informed consent for folder review was waived. Data from all included patients was anonymised prior to statistical analysis.

### A. Study population

A retrospective case cohort study was conducted at Charlotte Maxeke Johannesburg Academic Hospital (CMJAH), the only accredited public transplant facility in the greater Johannesburg area. The centre is the referral unit for all state hospitals within this region. A record review of all potential living donors assessed since the inception of the centre's living donor program in 1 January 1981 to 31 July 2015 was conducted. No exclusion criteria were applied.

Donor evaluation at this facility is in accordance with existing KDIGO guidelines. Owing to limitations in available immunosuppressive therapies, immunologically challenging procedures including ABO and HLA incompatible transplants are not routinely performed at the centre. The centre accepts medically complex donors in accordance with best practice guidelines, with the proviso that the risk of donation is deemed to be within acceptable limits by the attending multidisciplinary transplant team and that informed consent can be obtained from the donor. This subgroup includes donors with pre-existing hypertension where blood pressure (BP) is well-controlled on a single agent with no target organ damage and those with a body mass index (BMI) of 25–30 kg/m$^2$ prior to donation.

### B. Data collection

All parameters were captured across the study period with data being extracted from the clinical record filing system at the centre. Information gathered for all potential living donors included demographics, relation to the intended recipient and the outcome of the eligibility evaluation. If excluded from donation, the reason for non-donation was documented. For those eligible for living donation, clinical and laboratory parameters including BMI, BP, urine albumin excretion rate (AER) and serum creatinine were recorded at pre-donation work-up and follow-up. Any pre-existing medical conditions were noted. Hypertension following donation was defined as per documentation of either the diagnosis in medical records or the use of anti-hypertensive therapy. The AER was assessed by 24-hour urine collection or a spot urine albumin-creatinine ratio. Pre-donation glomerular filtration rate was determined using chromium-51-ethylene-diamine-tetra-aceticacid scans as standard protocol in the unit during the study period. Pre- and post-donation estimated glomerular filtration rates (eGFRs) were calculated using the Chronic Kidney Disease Epidemiology Collaboration (CKD-EPI) formula, at the time of this study the ethnicity correction factor was still in use for black donors. Access to genetic testing for apolipoprotein L1 (APOL-1) genotypes is not available in this unit. All laboratory and clinical parameters were performed at a single facility. Domicile in relation to the transplant centre, the number of post-donation follow-up visits and the last follow-up date were recorded. Post donation follow-up visits are routinely scheduled at six weeks, 3 months and then at six monthly intervals. If last follow-up was more than six months before the study end-point (31-07-2015), donor default was assumed. The reason for loss to

follow-up was recorded where known. Mortality data was ascertained based on centre knowledge of death as reported by next of kin.

## C. Statistical analysis

Distribution was assessed by the Shapiro Wilk W test and visual inspection of the nomogram. Continuous variables are expressed as means and standard deviations (SDs) or medians and interquartile ranges (IQRs), where distributions were Gaussian or non-Gaussian respectively. Categorical variables are presented as percentages. Statistical comparisons were performed with the Student's t-test for continuous normally distributed variables and the Pearson Chi-squared test for categorical variables. Where appropriate, the one-way ANOVA and Wilcoxon matched pairs testing were applied. For the successful donor cohort, a logistic regression model was performed to identify pre-donation factors associated with a reduced eGFR of $<60$ ml/min/1.73 m$^2$ at one-year post donation follow-up. A p-value of less than 0.05 was considered to indicate statistical significance. Statistical analyses were performed using Statistical for Windows version 12.0 (StatSoft Inc., 2015, Tulsa, OK, USA).

## Results

### A. Donor characteristics

1208 potential living donors were assessed. The demographic characteristics of all potential donors are presented in Table 1. Donors of Black African descent contributed the largest ethnic group ($n = 559$; 46%). Less than a quarter ($n = 298$; 24.7%) of potential donors eventually

**Table 1. Donor demographics.**

| Characteristic | Donors excluded n = 910 (75.3%) | Donors accepted n = 298 (24.7%) | Total evaluated n = 1208 |
|---|---|---|---|
| **Age (in years)** | | | |
| 18–21 | 54 (5.9)[1] | 9 (3.0) | 63 (5.2) |
| 22–29 | 213 (23.4) | 71 (23.8) | 284 (23.5) |
| 30–39 | 315 (34.6) | 113 (37.9) | 428 (35.4) |
| 40–49 | 221 (24.2) | 85 (28.5) | 306 (25.3) |
| 50–59 | 93 (10.2) | 19 (6.4) | 112 (9.3) |
| $\geq$60 | 14 (1.5) | 1 (0.3) | 15 (1.2) |
| **Gender** | | | |
| Female | 522 (57.4) | 175 (58.7) | 697 (57.7) |
| Male | 388 (42.6) | 123 (41.3) | 511 (42.3) |
| **Ethnicity** | | | |
| Black | 474 (52.1) | 85 (28.5) | 559 (46.2) |
| Caucasian | 358 (39.3) | 175 (58.7) | 533 (44.1) |
| Indian/Asian | 38 (4.2) | 26 (8.7) | 64 (5.3) |
| Mixed | 40 (4.4) | 12 (4.1) | 52 (4.3) |
| **Relation to recipient** | | | |
| Biological | 722 (79.0) | 269 (90.3) | 991 (82.0) |
| First degree relative | 631 (69.0) | 256 (85.9) | 887 (73.4) |
| Other relative | 91 (10.0) | 13 (4.7) | 104 (8.6) |
| Non-biological | 188 (21.0) | 29 (9.7) | 217 (18.0) |
| Directed | 180 (19.7) | 29 (9.7) | 209 (17.3) |
| Non-directed | 8 (0.9) | 0 | 8 (0.7) |

[1] Values are expressed as n (%)

underwent donor nephrectomy; there were thus 910 potential living donors who failed workup. LKD contributed a minority of transplants during the course of this series; in comparison, deceased donors accounted for 78% of all engraftments.

## B. Reasons for non-donation

Medical contraindications to donation were the most common cause for donor disqualification (*n* = 365; 40.1%, Table 2), of which obesity, hypertension, renal dysfunction, and HIV infection were most prevalent diagnoses. Psychosocial factors played a role in the exclusion of a further 209 donors, including 97 donors who voluntarily withdrew from the workup process, and 85 donors who did not return to complete assessment. Immunological barriers were cited in the exclusion of a further 173 potential donors. Thirty donors were excluded on the basis of radiologically diagnosed congenital renal anomalies or renovascular abnormalities, the latter including 15 patients with multiple renal vessels in which surgical approach to nephrectomy carried increased operative risk.

Outcome of donor evaluation varied significantly by ethnic group (Table 3). Failure to proceed to donor nephrectomy was more likely in Black donors (OR 2.72, 95% CI 2.05–3.62, p<0.001); black donors were also at increased odds of exclusion due to medical contraindications to donation (OR 1.57, 95% CI 1.23–2.00, p<0.001). A higher prevalence of hypertension and HIV infection were significant contributors to the exclusion of Black donors. A non-significant trend towards an increased rate of self-withdrawal from the evaluation process was observed for Black donors. ABOi was a more frequent immunological indication for exclusion in Black donors; preformed donor-specific HLA antibodies were, however, more significant as a barrier to donation in Caucasian donor-recipient pairs. A non-significant trend towards a higher frequency of unrelated donors was observed for non-black pairs (23.3% compared to 18.1% in Black African donors, p = 0.057); in related pairs, a higher frequency of second- and third-degree family relationships was observed for donors of Black African ethnicity (8.8% and 7.0% respectively compared to 6.6% and 3.3% for non-Black donors, p = 0.041).

LKD engraftments showed an increase after inception at this centre reaching a maximum of 22 transplants in 1995, following which a gradual decline in the LKD transplantation rate was noted, with a precipitous drop in the number of accepted donors occurring after 2006 (Fig 1).

## C. Outcomes following kidney donation

The median period of follow-up of donors was 3.7 years after donation (IQR 1.2–7.8 years). Clinical and laboratory parameters for this subgroup at baseline and at three successive post-donation follow-up points (at first visit, at one-year and at most recent visit) in donors with at least 5 years of follow-up are shown in Table 4. A progressive increase in measured blood pressure was observed at post-donation follow-up. Increase in systolic blood pressure between successive follow-up periods was significant in dependent sample t-testing between the first post-donation follow-up visit and follow-up at one year after donation (123.5 ± 17.8 mmHg and 128.2 ± 18.4 mmHg respectively, p = 0.012); using this methodology, significant increase in diastolic blood pressure was measured between pre-donation and first post-donation visits (73.1 ± 8.8 and 76.1 ± 12.1 mmHg respectively, p = 0.023). Thirty-eight donors (12.8%) had developed new-onset hypertension at the time of most recent follow-up. There was a significant increase in albuminuria over time between that measured pre-donation and that measured at the first post-donation follow-up visit (p<0.001 in Wilcoxon matched pairs testing), however the AER remained within normal limits. Thirteen donors (4%) developed an AER

**Table 2. Reasons for non-donation.**

| | n | % | | n | % | | n | % |
|---|---|---|---|---|---|---|---|---|
| **DONOR RELATED FACTORS** | 612 | 67.2 | MEDICAL | 365 | 40.1 | Hypertension | 90 | 9.9 |
| | | | | | | Obesity | 89 | 9.8 |
| | | | | | | Renal insufficiency[1] | 46 | 5.4 |
| | | | | | | HIV infection | 44 | 4.8 |
| | | | | | | Hypertension and obesity | 29 | 3.2 |
| | | | | | | Non-HIV active infections[2] | 20 | 2.2 |
| | | | | | | Haematological disorders[3] | 10 | 1.1 |
| | | | | | | Diabetes | 9 | 0.9 |
| | | | | | | Cardiovascular disease[4] | 7 | 0.7 |
| | | | | | | Non-diabetic endocrinopathy[5] | 4 | 0.4 |
| | | | | | | Pregnancy | 4 | 0.4 |
| | | | | | | Glomerular disease[6] | 3 | 0.3 |
| | | | | | | Rheumatological disease[7] | 3 | 0.3 |
| | | | | | | Advanced donor age | 3 | 0.3 |
| | | | | | | Neurological disease | 2 | 0.2 |
| | | | | | | Malignancy | 1 | 0.1 |
| | | | | | | Respiratory disease | 1 | 0.1 |
| | | | PSYCHOSOCIAL | 209 | 22.9 | Withdrew voluntarily | 97 | 10.7 |
| | | | | | | Lost to follow-up | 85 | 9.3 |
| | | | | | | Mental health disorder[8] | 14 | 1.5 |
| | | | | | | Substance abuse | 7 | 0.8 |
| | | | | | | Medico-legal exclusions[9] | 6 | 0.6 |
| | | | ANATOMICAL/RADIOLOGICAL | 30 | 3.3 | Multiple renal vessels | 15 | 1.7 |
| | | | | | | Congenital renal anomaly[10] | 8 | 0.9 |
| | | | | | | Atherosclerotic disease of non-renal arteries | 3 | 0.3 |
| | | | | | | Fibromuscular dysplasia | 3 | 0.3 |
| | | | | | | Renal artery stenosis | 1 | 0.1 |
| | | | UROLOGICAL | 8 | 0.9 | Nephrolithiasis | 6 | 0.7 |
| | | | | | | Obstructive uropathy | 2 | 0.2 |
| **DONOR–RECIPIENT FACTORS** | 173 | 19.0 | ABO incompatibility | | | | 96 | 10.5 |
| | | | Positive cytotoxic antibody crossmatch | | | | 77 | 8.5 |
| **RECIPIENT RELATED FACTORS** | 125 | 13.8 | RECIPIENT TRANSPLANTED | 52 | 5.7 | Donation from alternate living donor | 32 | 3.5 |
| | | | | | | Donation from deceased donor | 20 | 2.2 |
| | | | Candidate recipient demised during donor workup | | | | 47 | 5.2 |
| | | | Candidate recipient became medically ineligible for transplant | | | | 21 | 2.3 |
| | | | Candidate recipient withdrew voluntarily from programme | | | | 5 | 0.6 |

[1] Includes: Abnormality of renal function by Cr$^{51}$ EDTA or eGFR (41) and proteinuria (5);

[2] Includes: Hepatitis C (5), Hepatitis B (5), active *M. tuberculosis* (3), recurrent UTI (4), syphilis (2), and active CMV infection (1);

[3] Includes: persistent iron deficiency anaemia (7), persistent bicytopenia (1), Von Willebrand Disease (1), and bleeding diathesis (1);

[4] Includes: ischaemic heart disease (3), valvular heart disease (1), familial dyslipidaemia (1), chronic venous insufficiency (1), and cor pulmonale (1);

[5] Includes: primary hyperparathyroidism (2), hypothyroidism (1), and Graves' disease (1);

[6] Includes: Alport syndrome (2), active glomerulonephritis (1);

[7] Includes: Systemic lupus erythematosus with secondary antiphospholipid syndrome (1), primary antiphospholipid syndrome (1), and ankylosing spondylitis (1);

[8] Includes: bipolar mood disorder (1), psychotic disorder (1), major depressive disorder (3), and undifferentiated mental health disorders (9);

[9] Includes: inability to provide informed consent (3), donor incarceration (2), non-citizen donor (1);

[10] Includes: Unilateral hypoplastic kidney (3), autosomal dominant polycystic kidney disease (1), medullary sponge kidney (1), crossed fused renal ectopia (1), pelvic kidney with hydronephrosis (1), and supernumerary kidney (1)

**Table 3. Donor exclusion stratified by ethnicity.**

| | BLACK | NON-BLACK | p[1] | CAUCASIAN | p[2] |
|---|---|---|---|---|---|
| **OUTCOME OF DONOR EVALUATION** | | | | | |
| Successful donation | 85, 15.2% | 213, 32.8% | <0.001 | 175, 32.8% | <0.001 |
| Failed donation | 474, 84.8% | 436, 67.2% | | 358, 67.2% | |
| **INDICATIONS FOR DONOR EXCLUSION** | | | | | |
| All clinical exclusions[3] | 412, 82.9%[4] | 363, 63.0% | <0.001 | 297, 62.9% | <0.001 |
| Medical exclusions[5] | 206, 36.9% | 176, 27.1% | <0.001 | 139, 26.1% | <0.001 |
| Hypertension/renal insufficiency[6] | 96, 17.2% | 70, 10.8% | 0.001 | 54, 10.1% | 0.001 |
| Hypertension | 79, 14.1% | 45, 6.9% | <0.001 | 38, 7.1% | <0.001 |
| Decreased eGFR or CrCl | 18, 3.2% | 23, 3.5% | 0.757 | 14, 2.6% | 0.561 |
| Obesity[7] | 57, 10.2% | 61, 9.4% | 0.641 | 51, 9.6% | 0.728 |
| HIV | 41, 7.3% | 3, 0.5% | <0.001 | 1, 0.2% | <0.001 |
| Withdrew[8] | 96, 17.2% | 86, 13.3% | 0.06 | 70, 13.1% | 0.06 |
| Immunological | 94, 16.8% | 79, 12.1% | 0.02 | 69, 13.0% | 0.07 |
| ABO incompatibility | 67, 12.0% | 29, 4.5% | <0.001 | 24, 4.5% | 0.001 |
| Positive HLA cytotoxic antibody | 27, 4.8% | 50, 7.7% | 0.040 | 45, 8.4% | 0.020 |

[1] Black: Non-black donors, Pearson Chi-square test;

[2] Black: Caucasian donors, Pearson Chi-square test;

[3] Excluding recipient-related exclusions, donor pregnancy, donor medico legal contraindications, and exclusions based on technical difficulty (multiple renal vessels);

[4] Values are number of potential donors excluded, %;

[5] Excludes donor-recipient immunological barriers to donation and donor psychosocial factors;

[6] Includes all donors with hypertension, reduced eGFR or creatinine clearance, or proteinuria;

[7] Includes all obesity categories and exclusions for combination hypertension and obesity;

[8] Includes documented voluntary withdrawals and donors defaulting from the workup programme

of >300 mg/day, including one unrelated living donor who developed biopsy-proven idiopathic membranous glomerulonephritis eleven years after donation.

A significant decrease in eGFR was observed between pre-donation and at the first post-donation and one-year follow up-visits (p<0.001 for both follow-up measurements in Wilcoxon matched pairs testing); for those patients with at least 5 years of post-donation follow-up, there was no statistically significant difference between pre-donation eGFR and most recent follow-up measurement (p = 0.134). Eighty donors (27%) developed a eGFR of less than 60 ml/min/1.73 m$^2$ at most recent follow-up. However, none of the donors developed KF or

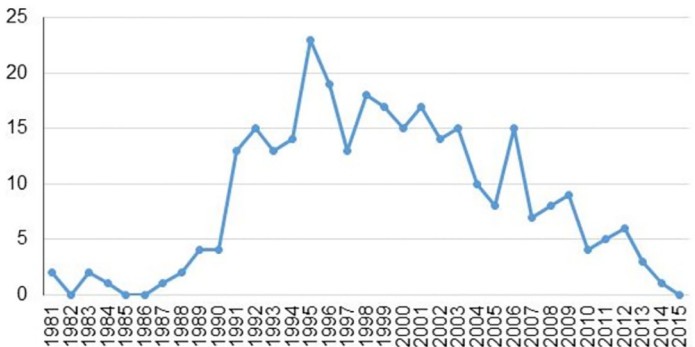

**Fig 1. Number of LKD engraftments by year.**

**Table 4. Comparative clinical parameters in successful living donors.**

| | Pre-donation | Post-donation | | | p[1] |
| --- | --- | --- | --- | --- | --- |
| | | First visit | At 1 year | At last visit[2] | |
| **AER (mg/d)[3]** | 4.0 (3.0–10.0) | 8.0 (3.0–23.0) | 6.0 (3.0–28.0) | 6.0 (3.0–27.0) | <0.001 |
| **Systolic BP (mmHg)[4]** | 118.5 ± 10.4 | 123.1 ± 17.4 | 128.2 ± 18.4 | 128.5 ± 18.1 | <0.001 |
| **Diastolic BP (mmHg)[4]** | 73.1 ± 8.8 | 77.2 ± 13.1 | 79.9 ± 13.3 | 81.5 ± 10.9 | <0.001 |
| **Creatinine (μmol/L)[4]** | 85.0 ± 15.1 | 114.9 ± 23.5 | 108.1 ± 23.7 | 98.1 ± 25.5 | <0.001 |
| **CKD-EPI eGFR (mL/min/1.73m²)[4]** | 93.4 ± 19.5 | 65.4 ± 17.6 | 68.9 ± 17.2 | 76.3 ± 22.3 | <0.001 |

[1] Friedman ANOVA by ranks;

[2] Median time to last visit 9.2 years, IQR 6.8–14.5 years, n = 104;

[3] Values are median (± IQR);

[4] Values are mean (± SD)

required KRT. Dependent t-testing revealed significant changes in eGFR measurement during the course of follow-up, with an expected decline in eGFR between pre- and first post-donation reading (93.0 ± 19.2 and 64.8 ± 17.1 ml/min/1.73m² respectively, p<0.001) followed by gradual improvement in eGFR between first post-donation follow-up and that at one year (65.9 ± 18.0 and 68.8 ± 17.2 ml/min/1.73m² respectively, p = 0.042), and between one-year and most recent visit in donors with at least 5 years of follow-up (68.9 ± 17.2 and 76.4 ± 22.6 ml/min/1.73m² respectively, p = 0.001). eGFR measurement at most recent follow-up in those donors with at least 5 years of follow-up was significantly lower than that measured in pre-donation workup (77.3 ± 22.9 and 93.4 ±19.6 ml/min/1.73m², p<0.001).

To accommodate for the effect of time after donation on eGFR measurement, regression analysis restricted to one-year follow-up was undertaken which identified multiple

**Table 5. Logistic regression analysis of pre-donation variables associated with a one-year post-donation eGFR<60 ml/min/1.73 m².**

| | Odds ratio | 95% CI | p |
| --- | --- | --- | --- |
| **Age at donation** | 1.04 | 1.02–1.04 | <0.001 |
| **Gender** | | | |
| Male | 1.90 | 1.70–2.13 | <0.001 |
| Female | 0.52 | 0.46–0.58 | <0.001 |
| **Ethnicity** | | | |
| Non-Black | 1.45 | 1.28–1.65 | <0.001 |
| Black African | 0.69 | 0.60–0.78 | <0.001 |
| **BMI (kg/m²)** | 1.03 | 1.02–1.05 | <0.001 |
| **[51] Cr-EDTA GFR (ml/min/1.73 m²)** | 0.99 | 0.99–1.00 | 0.040 |
| **CKD-EPI eGFR (ml/min/1.73 m²)** | 0.89 | 0.89–0.90 | <0.001 |
| **AER (mg/day)** | 1.02 | 1.01–1.02 | <0.001 |
| **Systolic hypertension[1]** | | | |
| Present | 2.59 | 2.07–3.23 | <0.001 |
| Absent | 0.39 | 0.31–0.48 | <0.001 |
| **Family history of hypertension or renal disease** | | | |
| Present | 1.53 | 1.36–1.72 | <0.001 |
| Absent | 0.65 | 0.58–0.73 | <0.001 |

[1] Denotes pre-existing hypertension well controlled on a single agent with no evidence of end organ damage

**Table 6. Donor ethnicity and parameters at most recent follow-up visit post donation.**

| | BLACK (n = 43) | NON-BLACK (n = 61) | p[1] | CAUCASIAN (n = 43) | p[2] |
|---|---|---|---|---|---|
| **Time since donation (years)** | 8.0 (6.4–13.3)[3] | 10.0 (7.0–15.8) | 0.060[4] | 10.6 (7.3–16.8) | 0.030 |
| **AER (mg/day)** | 4.0 (3.9–16.0) | 7.0 (3.0–28.0) | 0.430 | 10.0 (3.0–28.0) | 0.348 |
| **Systolic BP (mmHg)** | 128.3 ± 18.1[5] | 128.9 ± 18.3 | 0.881[6] | 132.9 ± 19.6 | 0.277 |
| **Diastolic BP (mmHg)** | 81.5 ± 11.8 | 82.0 ± 10.8 | 0.834 | 83.6 ± 11.0 | 0.408 |
| **Diagnosed hypertensive** | 19, 44.2%[7] | 25, 41.0%[8] | 0.745[8] | 20, 46.5% | 0.829 |
| **Creatinine (µmol/L)** | 95.5 ± 27.2 | 99.7 ± 24.1 | 0.418 | 102.1 ± 26.4 | 0.262 |
| **CKD-EPI eGFR (mL/min/1.73m$^2$)** | 86.7 ± 24.0 | 68.9 ± 17.5 | <0.001 | 66.3 ± 17.0 | <0.001 |
| **Presence of CKD[9]** | 9, 21.0% | 12, 19.7% | 0.875 | 8, 18.6% | 0.787 |
| **Presence of CKD and/or hypertension** | 19, 44.2% | 29, 48.3% | 0.677 | 19, 45.2% | 0.922 |

[1] Black versus Non-black;

[2] Black versus Caucasian;

[3] Values are median (IQR);

[4] Mann Whitney U test;

[5] Values are mean ± SD;

[6] Student t-test;

[7] Values are n, %;

[8] Pearson Chi-square test;

[9] eGFR <60ml/min/1.73m$^2$ and / or AER >30mg/day

pre-donation risk factors associated with reduced post-donation eGFR (Table 5). These included male gender, non-Black ethnicity, a family history of hypertension or renal disease and the presence of systolic hypertension prior to donation.

Black African ethnicity did not portend an increased risk of adverse outcome at most recent follow-up in those donors with at least 5 years of post-donation follow-up (Table 6). Renal function was better preserved in Black donors, although duration of post-donation follow-up was shorter in this group.

Sixty-seven donors (22.5%) had an isolated medical abnormality (IMA) prior to donation. This subgroup included donors with a pre-donation BMI of between 30–35 kg/m$^2$ ($n$ = 32); donors with a measured GFR of less than 80ml/min/1.73 m$^2$ ($n$ = 21) and those with pre-existing hypertension, well controlled on a single agent with no evidence of end-organ damage ($n$ = 14). A third of donors with an IMA were of Black African ethnicity. Class I obesity at baseline was not associated with an increased risk of developing hypertension (p = 0.090) or an eGFR of less than 60 ml/min/1.73 m$^2$ (p = 0.710) at follow-up. A pre-donation measured GFR of less than 80 ml/min/1.73 m$^2$ was not associated with an eGFR less than 60 ml/min/1.73 m$^2$ at most recent follow-up (p = 0.197). The incidence of new onset hypertension was not significantly greater in the IMA subgroup (p = 0.270).

With respect to non-renal outcomes, in the perioperative period sixteen donors (5.4%) suffered from complications including an iatrogenic pneumothorax ($n$ = 1); hospital acquired infection ($n$ = 3); and prolonged pain at the site of the nephrectomy scar ($n$ = 11). During the study period there were a total of 5 deaths (1.7%), one of which occurred as a result of significant haemorrhage in the immediate post-operative period. The remaining four deaths were due to trauma and unrelated to donation. Four donors (1.3%) required psychological support for post-donation major depressive disorder ($n$ = 2) and substance abuse ($n$ = 2).

## D. Donor follow-up

High rates of post-donation donor follow-up defaulting were observed in this study, with the median duration of follow-up being 4 years (IQR 1–8 years). At the time of study close (31 July 2015), only 59 donors (19.9%) remained in active follow-up. The median duration of follow-up in donors of Black African ethnicity (5.5 years, IQR 1.8–8.3 years) was longer than that in donors of other ethnic groups (2.6 years, IQR 0.9–7.6 years, p = 0.016); Caucasian donors had the shortest duration of follow-up (median 2.3 years, IQR 0.6–6.9 years). Domiciliary distance to the transplant centre did not affect follow-up duration. All donors aged 18–21 years (*n* = 9) at the time of donation were lost to follow-up. Gender and degree of relationship to the recipient were not associated with differences in follow-up.

## Discussion

Published data on living donation in Africa is limited, relying on small cohort studies [34–37]. In an analysis of 117 potential donors at a single centre in the Western Cape province of South Africa, the donor exclusion rate was 83% [34]. In the first South African study assessing outcomes of 33 Caucasian related living donors over thirty years ago, a rise in mean diastolic blood pressure with a tendency towards significant decline in creatinine clearance was noted at 5-year follow-up [35]. In a later series by Naicker et al., 135 donors with a similar demographic profile to the present study were assessed over a 10-year period; no significant difference in blood pressure or proteinuria was noted following donation, with a normal mean serum creatinine level over the follow-up period [36]. Findings of a subsequent study by Abdu et al. were similarly reassuring [37].

A substantial percentage of potential donors were excluded in the present study, with significant variation in the outcome of donor evaluation by ethnicity being observed. In a recent meta-analysis by Jagannathan et al., high risk APOL1 genotypes were associated with an increased risk for CKD in African Americans [38]. The potential presence of similar risk variants in Black South Africans may account for their disproportionate representation on national transplant waiting lists; registry data indicates that 52.2% of patients on KRT in South Africa are of Black ethnicity [8]. South Africa's socioeconomic disparities result in patients of the historically disadvantaged Black ethnic group being more likely to access KT through the public health system. These factors contribute to the predominance of patients of Black ethnicity amongst those receiving KRT in this sector and thus to the preponderance of potential donors in this study belonging to this group. It is however probable that simple demographics are insufficient to account for the increased odds of failed donation in Black potential donors. In this regard, medical contraindications to donation were significantly more frequent in Black donors, accounting for 36.9% of all exclusions in this group compared to 27.1% in Caucasians.

The screening of prospective donors frequently results in the incidental diagnosis of occult disorders. Consistent with previous studies, medical contraindications were the most common reason for non-donation in the present cohort, of which undiagnosed hypertension and obesity were most prevalent [17–20]. This reflects the current non-communicable disease burden in the South African population where hypertension and obesity rates are amongst the highest in sub-Saharan Africa, at 45% and 49% respectively; numerous studies support the role of distinct heritable factors in the prevalence of these disorders in populations of Black African ancestry [39–43]. Analyses of prospective African American donors have reported similar high exclusion rates on the basis of medical contraindications [20, 21, 44]. Weng et al. reported that candidates who are Black are less likely than Caucasians to successfully complete workup, with hypertension and obesity being the most common reasons for exclusion [21]. Recent advances

since the present study have shown that the inclusion of the race coefficient in estimating GFR by the CKI-EPI equation may underestimate the prevalence of CKD in Black individuals [45]. Since local protocol emphasises the use of isotope-measured GFR, the possibility of overestimation of renal function in pre-donation evaluation in this study is limited.

In contrast to reports from the developed world, the present study shows a significant contribution to donor exclusion by communicable diseases, primarily HIV infection [17, 20]. In 2018, the estimated seroprevalence rate of HIV in adult South Africans was 19%; with women of Black African descent being particularly affected [46]. Together, communicable and non-communicable diseases, reflective of epidemiological trends in the predominantly Black African population of South Africa, accounted for 42% of all donor exclusions in this study. Immunological barriers resulted in the exclusion of 19% of donor candidates. ABO incompatibility was an important factor in the exclusion of potential donors of Black African ethnicity. Although non-related donors were more common in non-Black donor pairs, higher-order degrees of relationship were more common in Black African pairs, likely reflecting extended kinship ties in this community; this may account for the higher frequency of ABO incompatibility in this group. Novel modalities including paired donor exchanges and desensitization protocols offer a means to accommodate such donors in a transplant programme [47, 48]. In developed settings, these approaches have increased the utilisation of living donors in otherwise incompatible donor-recipient pairs [49–51]. Creation of a national protocol for use in desensitization programmes across transplant centres in South Africa may further encourage safe donation across immunological barriers.

Twenty percent of otherwise medically suitable donors either withdrew voluntarily from donation (10.7%) or defaulted from the work-up process (9.3%); a non-significant trend towards increased withdrawal was observed in Black African donors. In a recent study by Kearney et al, the rate of withdrawal in Black potential donors was twofold higher as compared to Caucasian donors at a UK transplant centre [52]. Similar rates have been reported from multiple centres across the United States [20, 21]. African American donors have been shown to be more likely to decide against donation; reasons for this disparity in willingness to donate include cultural beliefs, perceived medical mistrust and socioeconomic inequality [20, 21, 53, 54]. It has been suggested that enhanced pre-evaluation education programs may reduce the occurrence of donor withdrawal [55]. In developed settings, cost-effective strategies including home visits and the use of social media outlets have proven beneficial in reducing racial disparities in living donor rates [56]. Such measures may be implementable in resource constrained settings to augment living donor pools.

Donation patterns in this study were comparable to that of developed societies. Females were the main contributors to the donor pool similar the United Kingdom where six in ten potential donors are female [57]. Female preponderance in potential donors was most significant among Black individuals as compared to other ethnic groups in this cohort. This potentially reflects the influence of sociodemographic and cultural determinants in transplantation, where women tend to assume a role of responsibility and self-sacrifice more frequently than male counterparts. There were no donations across ethnic groups in this cohort, denoting the deleterious legacy effect of South Africa's Apartheid past. Directed, non-biologically related donations occurred in compliance with transplant legislation in South Africa (Chapter 3 of the National Health Care Act 61 of 2003). Predominately for psychosocial contraindications, no potential altruistic donors proceeded to transplantation in this cohort. The rate of LKD engraftments in our centre has declined since 1995. The present study methodology does not permit full interrogation of this data, but the finding is consistent with trends reported for the broader South African context [5]. Moosa et al. have suggested that this decline is multifactorial in aetiology, with the emphasis placed on primary healthcare by the post-Apartheid

government having led to an erosion in the infrastructure and skills required by complex disciplines such as transplantation [5].

The presence of new onset hypertension and proteinuria following uninephrectomy are important clinical indicators of potential hyperfiltration injury to remnant glomeruli. As with previous studies, although a statistically significant rise in mean systolic blood pressure was observed following donation, this did not necessarily equate to clinical hypertension [58, 59]. In our cohort, thirty-eight donors (12.8%) had developed new-onset hypertension at the time of most-recent follow-up. The overall prevalence of hypertension in our donor population was however threefold lower than that of the general South African population [41]. Furthermore, AER at most recent follow-up showed a significant increase but remained within normal limits.

Although no donor required the institution of KRT, eighty (27%) demonstrated an eGFR less than 60 ml/min/1.73 $m^2$ at most recent follow-up. Pre-existing systolic hypertension (well controlled on a single agent with no organ damage) was associated with a two-fold greater risk of one-year eGFR below 60 ml/min/1.73 $m^2$. A family history of hypertension or renal disease was also associated with this outcome. These variables are possible indicators of the presence of underlying APOL1 risk variants or reduced nephron endowment respectively [60, 61]. The perioperative mortality rate of 0.3% in this study, comprising one death related to the surgical procedure with the remainder being of causes unrelated to donation, is comparable to that of centres in the United States [62].

Emerging prospective data from predominantly Caucasian donor cohorts report similar findings [63–65]. Janki et al. demonstrated excellent donor outcomes at between five and ten years of donor follow-up, with the incidence of hypertension amongst donors being comparable to the general Dutch population and no donors developing proteinuria [54, 65]. In addition, despite an initial expected decline in eGFR immediately following donation, renal function subsequently stabilised and no donor required the institution of KRT.

Although reassuring, it is unclear whether the findings of studies such as that of Janki et al. can be extrapolated to the African setting [64, 65]. Indeed, some data suggests a greater risk of CKD in African American donor subgroups [23–26, 66, 67]. In contrast to these studies, Black African donors in our cohort demonstrated superior renal outcomes compared to Caucasian counterparts, with Black African donors with at least 5 years of post-donation follow-up demonstrating a significantly higher eGFR than non-Black or Caucasian counterparts, and similar or non-significantly lower AER and blood pressure readings. These encouraging outcomes may reflect the application of a more conservative approach in the selection of Black African donors in the local setting as their post-donation risks have thus far been poorly defined.

Donor follow-up was poor during the course of the study. This remains a challenge globally as numerous international studies report similarly high lost to follow-up rates [68–70]. Our centre is the only state transplant facility in Johannesburg; follow-up visits thus necessitate that donors travel considerable distances at personal expense. Poor follow-up amongst young donors aged 18–21 years in our cohort may reflect psychological factors, including a lack of insight regarding the need for follow-up. At present, there are no established guidelines outlining the ideal duration and intervals for donor follow-up [71]. Given that close monitoring following donation affords the opportunity for early intervention should concerns arise, measures to ensure life-long follow-up of all kidney donors should be advocated [72–75].

There are limitations to this study. A relatively short median follow-up duration with a significant lost to follow-up rate may have led to adverse donor outcomes being underestimated as the onset of such typically occurs many years following donation. Conversely, adverse outcomes may have been overestimated in the event that symptomatic donors followed up more frequently. Comparative analysis with an appropriately matched control group of non-donors

in the South African population would be of benefit in adjusting for the effect of confounding factors on donor outcomes. Follow-up GFR was estimated using the CKD-EPI equation; ideally, if resource constraints were not a consideration, measured GFRs performed using radionuclide-based techniques would be preferable as a more accurate assessment of renal function in living donors. In addition, given the demographic profile of potential donors at our centre, genotyping for APOL1 alleles as a part of the pre-donation risk assessment may potentially be useful if resources allow [76, 77].

Strengths of this current study include a comprehensive assessment of living kidney donation in a developing country. As CMJAH is the only state facility that offers transplant services in the greater Johannesburg area, this study provides an accurate overview of living donor assessments and outcomes in this region over more than three decades. Data analysis at various points in donor follow-up allowed for the evaluation of adverse outcomes including the evolution of renal dysfunction following donation. The cohort size allowed comparison of work-up and post-donation outcomes between different ethnic subgroups. In addition, this study highlights possible ways to expand living donor pools in developing settings. These include population education regarding donation and addressing reasons for voluntary withdrawal amongst potential donors. Furthermore, circumventing immunological barriers to donation with the aid of desensitisation and the introduction of domino transplantation into state facilities is of importance.

In conclusion, this study adds to evidence supporting the ongoing practice of living kidney donation among carefully selected prospective donors. The present study is the largest single-centre report on living kidney donation in sub-Saharan Africa, and is particularly relevant due to the burgeoning demand for a sustainable form of definitive kidney replacement therapy in developing countries. In addition, analysis of this cohort does not appear to indicate adverse outcomes for donors of Black African descent.

## Supporting information

**S1 Data. Database key.**
(PDF)

**S1 Dataset.**
(XLSX)

## Acknowledgments

The authors gratefully acknowledge the assistance of Professor Saraladevi Naicker in the preparation of this manuscript.

## Author Contributions

**Conceptualization:** Chandni Dayal, Malcolm Davies, Nina Elisabeth Diana, Anthony Meyers.

**Data curation:** Chandni Dayal, Malcolm Davies, Nina Elisabeth Diana, Anthony Meyers.

**Formal analysis:** Chandni Dayal, Malcolm Davies, Nina Elisabeth Diana, Anthony Meyers.

**Investigation:** Chandni Dayal, Malcolm Davies.

**Methodology:** Chandni Dayal, Malcolm Davies, Nina Elisabeth Diana, Anthony Meyers.

**Resources:** Chandni Dayal.

**Supervision:** Malcolm Davies, Nina Elisabeth Diana, Anthony Meyers.

**Validation:** Malcolm Davies, Nina Elisabeth Diana, Anthony Meyers.

**Writing – original draft:** Chandni Dayal.

**Writing – review & editing:** Malcolm Davies, Nina Elisabeth Diana, Anthony Meyers.

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
