## [Decision Letter · Decision Letter 0]

24 Feb 2022

PONE-D-22-00868Living kidney donation in a developing countryPLOS ONE

Dear Dr. Dayal,

Thank you for submitting your manuscript to PLOS ONE. After careful consideration, we feel that it has merit but does not fully meet PLOS ONE’s publication criteria as it currently stands. Therefore, we invite you to submit a revised version of the manuscript that addresses the points raised during the review process.

This topic is of course massively important and will need our full support to see what can be learned from your experience in this regard. However, the reviewers have highlighted significant issues with regards to methodology and data analysis, and many items that were not mentioned in the paper that are important for the reader to understand. See detailed comments below. They will need to be thoroughly addressed in revisions and a point-by-point rebuttal. Please be aware that the revised MS will undergo vigorous re-review.

We look forward to receiving your revised manuscript.

Kind regards,

Frank JMF Dor, M.D., Ph.D., FEBS, FRCS

Academic Editor

PLOS ONE

Journal Requirements:

Reviewers' comments:

Reviewer's Responses to Questions

**Comments to the Author**

1. Is the manuscript technically sound, and do the data support the conclusions?

Reviewer #1: Yes

Reviewer #2: Partly

2. Has the statistical analysis been performed appropriately and rigorously? 

Reviewer #1: Yes

Reviewer #2: N/A

3. Have the authors made all data underlying the findings in their manuscript fully available?

Reviewer #1: Yes

Reviewer #2: Yes

4. Is the manuscript presented in an intelligible fashion and written in standard English?

Reviewer #1: Yes

Reviewer #2: No

5. Review Comments to the Author

Reviewer #1: Thanks to the authors for providing such clear insights on renal services and transplantation in South Africa.

1. According to the local registry, it is preferable to add data about South Africa's total population, approved transplant centers in the country, and the percentage of DD procedures per year.

2. In the methodology section, the authors need to add data about the annual transplant rate at the CMJAH (DD+LD) and whether they perform immunologically challenging (ABOi and HLAi) transplants?

3. Did all potential donors had a measured Ch-51- EDTA scan since 1983?

4. Do authors think that removing ethnicity correction factors for black donors will affect the donation decision? Better to comment on this in the discussion.

5. As in many other developing societies, still, females are the main contributor to the donor pool. Therefore, it is better to comment on this in discussion compared to other ethnic groups and regions.

6. Can authors mention donation between different ethnic groups (Black to Caucasian or vice versa) in the non biologically related donors?

7. Hypertension was the main factor for donor exclusion, but authors accepted that donors with stage I preexisting hypertension had no organ damage and controlled with one medication! This needs to be elucidated more clearly in the methodology section. The discussion should explain this as preexisting hypertension associated with OR 2.59 for one-year eGFR <60 ml/min.

This is applicable for those with a BMI of 30-35 kg/m2.

Authors should justify the selection of such donors by guidelines and their local practice and setting.

8. In the methodology section, the authors should mention whether genetic testing for APOL1 genotype is available or not?

9. I wonder if there is a current desensitization program or PKD in South Africa. Better to be mentioned.

Reviewer #2: I would like to congratulate the authors for their efforts to bring out this rare data and brought up the topic of challenges at a public kidney transplantation program in Africa. Especially most patients depending on public health support don’t have access to Dialysis that means sacrifice of so many patients that does not have access to hemodialysis. Can you please make a short comment about why peritoneal dialysis is not a depending choice of RRT in Africa as it happens in Mexico because of its economic benefits?

What is the reason to limit the study period in between 1983 and 2015? What is the number of transplants after 2015? Does the period after 2015 had any change in clinical practice or legislations in regard to living donation? Please comment on this issue.

The living donors includes a group of people who are non-biological couples(n=188) (Table 1). What is the legislation for these non-biological donors and none of the donors at non-directed group is accepted? Can you give a little more detail about these donors?

96 Patients had ABO incompatibility and excluded from pretransplant work up immediately. Inclusion of this group into the series may cause statistical bias because they were never evaluated medically or psychologically. What is your comment about this issue?

It is not clear how many donors are lost to fallow up out of 298 donors after living donation. The distribution of living donors over 32 years are important to mention as well. There are 104 donors (Black+ Non-Black) who have completed 5 years fallow up. This is a significant lost to fallow to make accurate statistical evaluation. The retrospective design of study is an important factor effecting the statistical significance as well.

I would like to congratulate the authors for sharing this series which is a tremendous work at the public hospital and brought up successful results regarding donor selection. I am aware of the hard clinical conditions and difficulties to bring out the retrospective data. The data collection is spread in between 27 years and the missing data is significant as well, especially about fallow up of the living donors.

6. PLOS authors have the option to publish the peer review history of their article (what does this mean?). If published, this will include your full peer review and any attached files.

Reviewer #1: No

Reviewer #2: **Yes: **Emin Baris Akin

---

## [Author Response · Author response to Decision Letter 0]

27 Mar 2022

4. Response to Reviewer comments 

Reviewer 1

1. According to the local registry, it is preferable to add data about South Africa's total population, approved transplant centers in the country, and the percentage of DD procedures per year.

Data pertaining to the total South African population size, accredited transplant centers and percentage of deceased donor procedures per year in South Africa has been added to the Introduction section of the manuscript. This data has been referenced from Statistics South Africa as well as the most recent published report by the South African Renal Registry. 

2. In the methodology section, the authors need to add data about the annual transplant rate at the CMJAH (DD+LD) and whether they perform immunologically challenging (ABOi and HLAi) transplants?

The hospital is a state sector facility; cost and limited access to the immunosuppressive therapies required in desensitization protocols remain prohibitive factors in the performance of immunologically challenging transplants at the center. 

Data regarding the overall transplant rate at CMJAH was obtained from available unit records and the Solid Organ Transplant Governance for the Gauteng Province. The following were noted:

i. No data is available on the number of deceased donor transplants performed at CMJAH from 1983-1991. 

ii. From 1991-2015 (the end of the study period), CMJAH performed a total of 1049 deceased kidney donor transplants. In the comparative time frame, the unit performed 293 living kidney donor transplants. Approximately 80% of donor kidneys were therefore procured from deceased donors for the study period. The center’s preponderance for deceased donor transplants is in line with national data and underscores the underutilization of living kidney donors in expanding the transplant pool.

The above information has now been included in the Methods and Results sections of the manuscript. A figure depicting the number of living donor engraftments by year at the center has also been added.

3. Did all potential donors have a measured Ch-51- EDTA scan since 1983?

The Department of Nuclear Medicine assists with Ch-51-EDTA scans at CMJAH. This service became available in 1988. There were five accepted donors in the cohort that did not have Ch-51-EDTA scans done at the facility (two in 1981, two in 1983 and one in 1987). The records of Ch-51-EDTA scans could not be traced for a further seven donors.

4. Do authors think that removing ethnicity correction factors for black donors will affect the donation decision? Better to comment on this in the discussion.

The authors acknowledge the limitations of computing estimated GFR using the Chronic Kidney Disease Epidemiology Collaboration (CKD-EPI) equation with and without its race coefficient for potential donors of Black ethnicity. Removal of race may have increased the prevalence of CKD in this subgroup and resulted in a greater proportion of Black kidney donor candidates being disqualified on the basis of an estimated GFR of <60 ml/min/1.73m2. 

To prevent the limitations of calculated GFRs from influencing the donation decision, as a part of the medical eligibility work-up, all potential donors at the institution have measured GFRs performed by Ch-51-EDTA scan. In this study, the result of the 51-Ch-EDTA GFR was documented where the potential donor was excluded on the basis of renal dysfunction. 

This important issue has now been raised in the discussion section of the manuscript. 

5. As in many other developing societies, still, females are the main contributor to the donor pool. Therefore, it is better to comment on this in discussion compared to other ethnic groups and regions.

The authors agree that the female preponderance of potential donors in this cohort is similar to findings in other societies. A comment on the influence of sociocultural factors in transplantation across various ethnic groups and regions has now been included in the discussion section. 

6. Can authors mention donation between different ethnic groups (Black to Caucasian or vice versa) in the non-biologically related donors?

No transplants between different ethnic groups occurred in non-biologically related potential or accepted donor-recipient pairs in this cohort. The authors acknowledge that this may reflect cultural determinants in transplantation as well as the legacy of South Africa’s Apartheid past. This has been included in the additional paragraph on donation patterns in the discussion section. 

7. Hypertension was the main factor for donor exclusion, but authors accepted that donors with stage I preexisting hypertension had no organ damage and controlled with one medication! This needs to be elucidated more clearly in the methodology section. The discussion should explain this as preexisting hypertension associated with OR 2.59 for one-year eGFR <60 ml/min. This is applicable for those with a BMI of 30-35 kg/m2. Authors should justify the selection of such donors by guidelines and their local practice and setting.

The chronic shortage of donor allografts in the local context has led to the expansion of the selection criteria for living donors at CMJAH. In accordance with best practice KDIGO guidelines, the center accepts marginal donors in the event that the attending multidisciplinary transplant team deems the risk of donation to be within acceptable limits and informed consent can be provided by the donor. Medically complex donors in this cohort included those with pre-existing hypertension well controlled on a single agent with no organ damage and those with a BMI of 30-35kg/m2. For further clarity, this has been included in the methodology section and revised in the discussion. 

8. In the methodology section, the authors should mention whether genetic testing for APOL1 genotype is available or not?

Genetic testing for high-risk APOL1 genotypes would be invaluable in informing donation risk for Black potential donors in particular. Unfortunately, cost and limited access to specialized laboratory services in a resource constrained South African setting restricts widespread clinical use of this test. This has been added to the methodology section and is also included in the discussion on limitations of the study in the manuscript. 

9. I wonder if there is a current desensitization program for PKD in South Africa. Better to be mentioned.

While centre-specific desentization protocols are applied at various transplant facilities in the country, a standardized, nationally endorsed guideline for desensitization is yet to be formulated for use in South Africa. 

A review of current data on desensitization as well as engagement between transplant facilities across the country may facilitate the inception of a guideline for widespread use in resource constrained transplant programs. This may encourage more consistent transplantation across immunological barriers at a national level. 

The above has now been included in the discussion section of the manuscript. 

Reviewer 2

1. Especially most patients depending on public health support don’t have access to dialysis that means sacrifice of so many patients that does not have access to hemodialysis. Can you please make a short comment about why peritoneal dialysis is not a depending choice of RRT in Africa as it happens in Mexico because of its economic benefits?

The authors acknowledge the underutilization of peritoneal dialysis as a bridge to kidney transplantation in developing settings. South African Renal Registry Data consistently highlights hemodialysis as the predominant kidney replacement modality across both the public and private sectors in the country. This occurs in spite of the ‘Peritoneal Dialysis First’ initiative at most public sector facilities. 

The challenges limiting widespread use of PD in the South African setting are common to other parts of Africa and include:

i. Sociodemographic factors, such as difficulty in transportation of consumables to rural settings, inadequate water and sanitation facilities, low electrification rates and unsuitable living circumstances

ii. Healthcare system related factors, such as cost limitations and a shortage of trained PD nurses and nephrologists to supervise therapy

iii. Patient related factors, such as burnout and the perception of PD as being an inferior kidney replacement modality as compared to hemodialysis 

A comment pertaining to socioeconomic challenges limiting access to all forms of kidney replacement therapy, including peritoneal dialysis, in developing countries has been added to the introduction section of the manuscript. 

2. What is the reason to limit the study period in between 1983 and 2015? What is the number of transplants after 2015? Does the period after 2015 had any change in clinical practice or legislations in regard to living donation? Please comment on this issue.

The present manuscript has arisen from data collected for the principal investigators Masters dissertation in 2015. The end of the study period therefore reflects when the protocol for the dissertation was submitted and approval for data collection was obtained from the institutions’ ethics committee.

It does not reflect any change in clinical practice or legislation regarding living kidney donation at the institution. 

There were a total of 153 kidney transplants at the center from 2015 to present, of which only 21 were procured from directed living kidney donors. The remainder were all deceased donor transplants. 

3. The living donors includes a group of people who are non-biological couples(n=188) (Table 1). What is the legislation for these non-biological donors and none of the donors at non-directed group is accepted? Can you give a little more detail about these donors?

Legislation regarding non-biological living donors in South Africa is guided by the National Health Act 61 of 2003. Chapter 3 of this regulation stipulates that all non-biological living donors must provide informed consent for transplantation. Furthermore, a written application containing specific clinical information regarding the prospective donor-recipient pair must be submitted to the Minister of Health (as highlighted in annexure C of the National Heath Act 61). Upon review of this information, written Ministerial permission must be provided to the transplant center before unrelated living donation may proceed. This legislative process was adhered to in the present study. 

As noted in table 1, no altruistic potential donors successfully proceeded to kidney transplantation in this cohort. In this subgroup of eight potential donors, six were disqualified for psychosocial reasons including voluntary withdrawal, expectation of monetary remuneration and loss to follow-up. A further two were excluded for medical reasons which were refractory hypertension and morbid obesity. 

A comment on the above has now been included in the additional paragraph on donation patterns in the discussion section of the manuscript. 

4. 96 Patients had ABO incompatibility and excluded from pre-transplant work up immediately. Inclusion of this group into the series may cause statistical bias because they were never evaluated medically or psychologically. What is your comment about this issue?

The authors acknowledge the potential introduction of statistical bias with the inclusion of immunological mismatch as the reason for disqualification from donation as this subgroup may have included individuals with other additional medical or psychological reasons that preclude donation. However, since local practice limits donation in the setting of ABO or HLA incompatibility, immunological factors constitute an important factor in limiting LKD and the inclusion of this category in the study population provides insight on methods to expand LKD, in line with the stated objectives of the study.

5. It is not clear how many donors are lost to follow up out of 298 donors after living donation. The distribution of living donors over 32 years are important to mention as well. There are 104 donors (Black+ Non-Black) who have completed 5 years follow up. This is a significant lost to follow up to make accurate statistical evaluation. The retrospective design of study is an important factor effecting the statistical significance as well.

Results have been expanded to include a better representation of donor follow-up. We have included the median duration of follow-up for all donors and indicated the number of donors remaining on active follow-up at study close. We have also acknowledged the limitations arising from the significant rate of post-donation loss to follow-up in the discussion. Follow-up drop-out after donation is an important concern in determining the safety of donation in our population and we have highlighted this in the discussion.

---

## [Decision Letter · Decision Letter 1]

25 Apr 2022

Living kidney donation in a developing country

PONE-D-22-00868R1

Dear Dr. Dayal,

We’re pleased to inform you that your manuscript has been judged scientifically suitable for publication and will be formally accepted for publication once it meets all outstanding technical requirements.

Kind regards,

Frank JMF Dor, M.D., Ph.D., FEBS, FRCS

Academic Editor

PLOS ONE

Additional Editor Comments (optional):

Reviewers' comments:

Reviewer's Responses to Questions

**Comments to the Author**

1. If the authors have adequately addressed your comments raised in a previous round of review and you feel that this manuscript is now acceptable for publication, you may indicate that here to bypass the “Comments to the Author” section, enter your conflict of interest statement in the “Confidential to Editor” section, and submit your "Accept" recommendation.

Reviewer #1: All comments have been addressed

Reviewer #2: (No Response)

2. Is the manuscript technically sound, and do the data support the conclusions?

Reviewer #1: Yes

Reviewer #2: Yes

3. Has the statistical analysis been performed appropriately and rigorously? 

Reviewer #1: Yes

Reviewer #2: Yes

4. Have the authors made all data underlying the findings in their manuscript fully available?

Reviewer #1: Yes

Reviewer #2: Yes

5. Is the manuscript presented in an intelligible fashion and written in standard English?

Reviewer #1: Yes

Reviewer #2: Yes

6. Review Comments to the Author

Reviewer #1: (No Response)

Reviewer #2: (No Response)

7. PLOS authors have the option to publish the peer review history of their article (what does this mean?). If published, this will include your full peer review and any attached files.

Reviewer #1: No

Reviewer #2: **Yes: **Emin Baris Akin

---

## [Editor Report · Acceptance letter]

29 Apr 2022

PONE-D-22-00868R1 

Living kidney donation in a developing country 

Dear Dr. Dayal:

I'm pleased to inform you that your manuscript has been deemed suitable for publication in PLOS ONE. Congratulations! Your manuscript is now with our production department. 

Kind regards, 

on behalf of

Dr. Frank JMF Dor 

Academic Editor

PLOS ONE